**DOI: 10.1038/ncomms13868**　　**OPEN**

# Electronic single-molecule identification of carbohydrate isomers by recognition tunnelling

JongOne Im[1,2], Sovan Biswas[1,3], Hao Liu[1,3], Yanan Zhao[1], Suman Sen[1,3], Sudipta Biswas[1,3], Brian Ashcroft[1], Chad Borges[1,3], Xu Wang[3], Stuart Lindsay[1,2,3] & Peiming Zhang[1]

Carbohydrates are one of the four main building blocks of life, and are categorized as monosaccharides (sugars), oligosaccharides and polysaccharides. Each sugar can exist in two alternative anomers (in which a hydroxy group at C-1 takes different orientations) and each pair of sugars can form different epimers (isomers around the stereocentres connecting the sugars). This leads to a vast combinatorial complexity, intractable to mass spectrometry and requiring large amounts of sample for NMR characterization. Combining measurements of collision cross section with mass spectrometry (IM–MS) helps, but many isomers are still difficult to separate. Here, we show that recognition tunnelling (RT) can classify many anomers and epimers via the current fluctuations they produce when captured in a tunnel junction functionalized with recognition molecules. Most importantly, RT is a nanoscale technique utilizing sub-picomole quantities of analyte. If integrated into a nanopore, RT would provide a unique approach to sequencing linear polysaccharides.

[1] Biodesign Institute, Arizonan State University, Tempe, Arizona 85287, USA. [2] Department of Physics, Arizonan State University, Tempe, Arizona 85287, USA. [3] School of Molecular Sciences, Arizonan State University, Tempe, Arizona 85287, USA. Correspondence and requests for materials should be addressed to S.L. (email: Stuart.Lindsay@asu.edu) or to P.Z. (email: Peiming.Zhang@asu.edu).

Carbohydrates, particularly those glycosylating proteins and lipids (glycans), play an essential role in biological processes at all levels, such as protein folding[1], cell adhesion[2], signal transduction[3], pathogen recognition[4] and immune responses[5]. Over 50% of all human proteins are glycosylated[6], while aberrant glycosylation of proteins is associated with oncogenic transformation[7]. Since many carbohydrates are epimers, anomers and/or regioisomers, mass spectrometry cannot identify those sharing a molecular weight without additional chemical steps[8]. The problem has recently been addressed by combining ion-mobility spectrometry, which uses collision cross-sections, with mass spectrometry (IM–MS) to separate isomers[9], but IM–MS has not been able to resolve closely related epimers because they have almost identical collision cross-sections.

Here, we introduce an electron tunnelling technique as a new tool to identify carbohydrates at a single-molecule level. It is capable of analysing nanomolar concentrations in volumes of a few microliters, using less than a picomole of sample. It counts the number of individual molecules in each subset in a population of coexisting isomers, and is quantitative over more than four orders of magnitude of concentration. It resolves epimers that are not well separated by ion-mobility, and can detect glycosylation of a peptide. A tunnel junction integrated into a solid-state nanopore has the potential to sequence linear oligo- and poly-saccharides.

## Results

**Recognition Tunnelling**. Recognition tunnelling (RT) generates characteristic electron tunnel-current spikes, when an individual analyte is trapped in a nanogap by capture molecules tethered to two electrodes separated by a few nanometres. It has been used to identify individual nucleobases, amino acids and peptides at a single-molecule level[10–14]. In the present study of carbohydrates, the capture molecule was 4(5)-(2-mercaptoethyl)-1H-imidazole-2-carboxamide (ICA, Fig. 1a)[15]. ICA exists in an equilibrium of two tautomeric forms, bearing multiple hydrogen bond donors and acceptors for recognition and a two-carbon alkyl chain terminated with a thiol function for attachment to electrodes. Its small size allows for formation of a hydrogen bonded triplet complex with each of monomers constituting biopolymers in the tunnel gap. We first applied RT to distinguish between two anomeric isomers, namely methyl α-D-glucopyranoside (α-$^M$Glu) and methyl β-D-glucopyranoside (β-$^M$Glu). They are non-reducing monosaccharides, differing only in the relative orientation of the methoxy group (marked in red), as shown in Fig. 1b,c. These two compounds were purchased from a commercial source (Supplementary Methods) and confirmed by NOESY NMR for their correct configurations before RT measurements (Supplementary Methods and Supplementary Fig. 1). Computer simulations (Supplementary Methods) show that each of them can be appropriately accommodated in a 2.2 nm wide tunnel gap by hydrogen bonding to a pair of ICA molecules, and that the two molecules can be distinguished by their hydrogen-bonding patterns and energies in principle. As shown in Fig. 1d,e, α-$^M$Glu forms two hydrogen bonds with each ICA, resulting in a more stable complex than that of β-$^M$Glu, which interacts with each ICA through a single hydrogen bond. We have observed 1:1 and 2:1 complexes formed between ICA and these two anomers by analysing aqueous solutions of ICA mixed with the analytes in a 2:1 ratio using electrospray ionization mass spectrometry (ESI-MS), a method that provides a snapshot of solution equilibria in the gas phase[16] (Supplementary Tables 2–4 and Supplementary Methods; MS data for the other monosaccharides and disaccharides used in the

present work are also included there). The RT measurements were carried out using a scanning tunnelling microscope (STM), following a process of functionalizing palladium (Pd) STM probes and Pd-substrates with ICA in ethanolic solutions (see Methods for details of functionalization and characterization of probes and substrates), mounting them in a PicoSPM instrument (Agilent), recording a RT spectrum of the phosphate buffer solution, pH 7.4 to assure that a clean baseline is achieved (see Methods, Supplementary Fig. 4; Supplementary Table 6 for the control experiments), introducing an analyte solution (typically 100 μM in a 1.0 mM phosphate buffer, pH 7.4) to the liquid cell, and collecting current recordings with a tip-to-substrate bias of 0.5 V (see Methods for experimental details). For each analyte, four sets of RT data were collected from four different runs, each of which was measured with a freshly made probe and substrate. Examples of the tunnel-current signals generated by α-$^M$Glu and β-$^M$Glu are shown in Fig. 1f,g. They consist of a train of current spikes similar to those theoretically predicted for other hydrogen-bonded structures[17]. Driven by thermal fluctuations, the signals are stochastic but rich in information about the bonding, and thus the identity, of the trapped analyte molecule.

**Signal Analysis for α-$^M$Glu and β-$^M$Glu**. Signals appear in bursts, and strong correlations between signal features in a given cluster (see Data Analysis in Methods) suggest that each cluster reflects structural fluctuations within one particular capture geometry for one molecule. Average count rates (Supplementary Table 7) are much lower than the count rate within a signal cluster, so the number of distinct molecules sampled in a given run is likely much smaller than the total number of peaks measured. While the lifetime of the bonded complex may be seconds, small fluctuations in bonding geometry give rise to rapid on–off (telegraph) switching because of the exponential dependence of tunnel current on geometry[17]. Individual signal features of these two anomers are broadly distributed and overlapped, as shown in Fig. 1h,i. However, when a three-dimensional probability density plot is made of the pairs of values of signal features that occur together in each signal spike (Fig. 1j), the distributions separate. Here we have chosen two principle component vectors, constructed as described in Supplementary Methods, with the third dimension (the frequency) represented by brightness. The colours that represent the analytes (red, α-$^M$Glu, green, β-$^M$Glu) are mixed so regions of overlap are yellow. The two principle component vectors (SI section 8) used in Fig. 1j were chosen to demonstrate the underlying mechanism of a much more accurate analysis that can be achieved by including many more signal features in a multidimensional distribution using a Support Vector Machine, SVM, a machine-learning algorithm[14], the details of which are described in our previous publication[18], and elaborated in the data analysis section of Methods.

In brief, an SVM was first trained using a randomly-selected 10% subset of data, based on up to 264 available signal features (see feature selection, Methods)[18]. This feature set is iteratively reduced to leave the smallest number of signal features that give a high accuracy for classifying the data, as tested on the remaining 90% of the data not used in training. The SVM first identified events that were common to data obtained from different samples owing to contamination, capture events that were insensitive to chemical variation and noise spikes generated by the STM electronics and servo control. These amounted to about 50% of all signal spikes. The remaining signals could be classified by SVM with accuracy of >99% for the two anomers (column a in Panel i, Table 1). Training on a subset of all four data sets (collected with four microscopically-different tunnel junctions) sets an upper

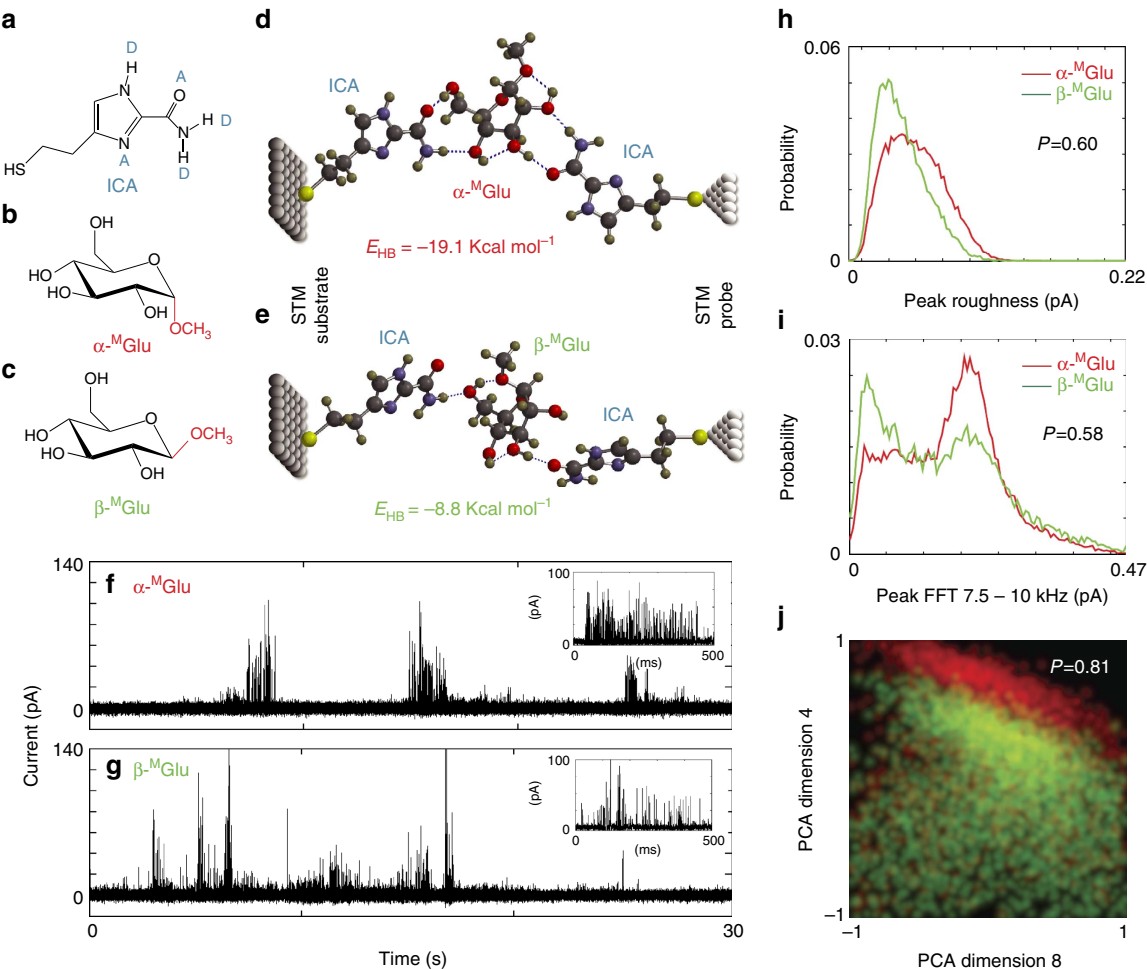

**Figure 1 | Recognition tunnelling analysis of anomers of methyl D-glucopyranoside. (a)** The recognition molecule ICA bearing a thiol linkage for bonding to metal electrodes, as well as a number of hydrogen bonding donors (D) and acceptors (A), through which a large range of analytes can be captured by a diversity of spatial arrangements resulting from tautomerism and rotation about σ-bonds; **(b)** Structure of α-$^M$Glu and **(c)** Structure of β-$^M$Glu, both of which can form hydrogen-bonded triplets with ICA molecules spanning a tunnel gap of 2.2 nm, as shown by computer simulations in **d,e**. Evidence of these complexes is provided by the current-spikes that appear only after an analyte solution is added to pure buffer solution in a tunnel gap (**f,g**). Distributions of signal features are broad and overlapped (red = α-$^M$Glu, green = β-$^M$Glu) as shown here for the peak roughness (s.d. of points above half maximum current) **h** and one frequency band in the Fourier transform of each peak (peak FFT 7.5–10 kHz, **i**) Data can be assigned to one analyte or the other with a probability (0.6, 0.58) only marginally above random, 0.5 (see Methods for details of the signal analysis). However, when the frequency with which multiple parameter values occur together is plotted (**j**) the accuracy with which data can be assigned increases to 80%. The plot shows the distribution of the simultaneous occurrence of two principle components, vectors composed of multiple parameter values as described in Supplementary Methods. When the distribution of parameter values is constructed in higher dimensions, separation increases to ~99%. This accuracy can be improved to ~99% using additional signal features. Colours in **j** are mixed so that overlapped points are yellow.

limit on accuracy (called the 'optimistic' accuracy). Practically, it can be achieved by calibrating the RT device with a standard solution before testing the sample to be analysed. More interesting is the 'predictive' accuracy, which was generated by training the SVM on data from three tunnel junctions and then using it to analyse data from the fourth junction. For example, both α-$^M$Glu and β-$^M$Glu were identified with predictive accuracies of 85.8% and 86.9%, respectively (column b in Panel i, Table 1). With 1 μM concentration, however, we obtain 94.1 and 94.8% predictive accuracy respectively for α-$^M$Glu and β-$^M$Glu (without compromising their optimistic accuracy, 99.6% for α-$^M$Glu and 98.8% for β-$^M$Glu). The accuracy is substantially improved at 1.0 μM presumably because multi-molecule binding events are less frequent at the lower concentration. This illustrates the potential of the RT technique to identify components in an unknown samples that have been previously characterized in a RT database.

**Analysis of data for other carbohydrate stereoisomers.** We extended the RT technique to measure other carbohydrate stereoisomers (see Supplementary Fig. 5 for their representative RT spectra). As summarized in Table 1, we have been able to distinguish glucose, glucosamine and N-acetylgluosamine from their respective C-4 epimers with ~99% optimistic accuracy (Panel ii, iii, iv, Table 1), and D-glucuronic acid from its C-5 epimer L-iduronic acid with ~98% optimistic accuracy (Panel v, Table 1). Although the predictive accuracy varied from 82.5 to 98.8%, at least one analyte in each of these pairs can be identified with the predictive accuracy of >90%, for each single signal spike. RT measurement is a stochastic process so its accuracy can be further improved by simply measuring the same sample multiple times. Note that these monosaccharides are reducing sugars so that each of them exists with an equilibrium distribution of anomeric isomers and a possible open-chain form in aqueous solution. NMR data suggest that the equilibrium would

**Table 1 | Accuracy of classifying carbohydrate pairs by SVM analysis of RT data.**

| Analyte | Accuracy (%) | | Analyte | Accuracy (%) | |
|---|---|---|---|---|---|
| | **a** Optimistic | **b** Predictive | | **a** Optimistic | **b** Predictive |
| i D-α-^MGlu | 99.8 | 85.8 | ii D-Glucose | 97.2 | 90.9 |
| D-β-^MGlu | 99.2 | 86.9 | D-Galactose | 99.5 | 85.8 |
| iii D-Glucosamine | 99.4 | 97.1 | iv D-N-Acetyl-Glucosamine | 99.3 | 96.3 |
| D-Galactosamine | 98.3 | 82.5 | D-N-Acetyl-Galactosamine | 99.5 | 89.3 |
| v D-Glucuronic Acid | 98.7 | 98.8 | vi α-D-Glucopyranosyl-(1→4)-D-glucopyranose (Maltose) | 99.5 | 88.5 |
| L-Iduronic Acid | 97.6 | 95.2 | β-D-glucopyranosyl-(1→4)-D-glucopyranose (Cellobiose) | 99.8 | 86.4 |
| vii β-Δ^{4,5}-D-UA(1→3)-D-GalNAc-4-O-Sulfate (D0A4)* | 98.8 | 85.1 | viii Ala-Ser-Ala-NH₂ | 96.8 | 89.1 |
| β-Δ^{4,5}-D-UA(1→3)-D-GalNAc-6-O-Sulfate (D0A6)* | 99.6 | 80.6 | Ala-Ser(GalNAc)-Ala-NH₂ | 94.6 | 96.2 |

*Lawrence disaccharide nomenclature used to indicate sulfation position[29].
The concentration of each analyte is about 100 µM.

not be affected by the presence of ICA (Supplementary Methods and Supplementary Fig. 7). So these epimers could be accurately identified because the training set for each epimer contained the same mixture of anomers. In addition, we were also able to distinguish with similar accuracy between two anomeric isomers of a disaccharide, maltose (α-D-glucopyranosyl-(1→4)-D-glucopyranose) and cellobiose (β-D-glucopyranosyl-(1→4)-D-glucopyranose) (Panel vi, Table 1), and between two regioisomers, D0A4 and D0A6, of a chondroitin sulphate (Panel vii, Table 1 and see Supplementary Methods for their synthesis), a repeating disaccharide unit of glycosaminoglycans (GAGs). We also used RT to distinguish between a peptide and its glycosylated form. As shown in Panel viii of Table 1, RT identified α-N-acetyl-galactosamine (GalNAc) linked to serine (a tumour biomarker, known as Tn antigen[19]) of a tripeptide and

the parent peptide from pooled data with a predictive accuracy of 96.2 and 89.1%, respectively.

Strikingly, RT was capable of identifying many different monosaccharides from pooled data with a high degree of accuracy. We built a data pool containing all the RT signals generated from 10 of the most abundant mammalian monosaccharides[20]. As shown in Table 2, the SVM was able to identify each of them with optimistic accuracy of ranging from 91 to 98% (random would be 10%). This indicates that RT can potentially be used in identification of carbohydrates in a wide glycospace. In contrast, IM–MS is ineffective in discriminating between galactose and mannose[21] or between glucose and galactose[9].

**Quantification of analytes.** We have determined the affinity of α-^MGlu binding to ICAs in the tunnel gap using RT. Figure 2 is a

**Table 2 | Accuracy of determining individual carbohydrates from a pool of RT data.**

(GlcNAc)
94.4

(Gal)
95.3

(Man)
95.6

(Neu5Ac)*
97.3

(L-Fuc)
91.7

(GalNAc)
96.4

(Glc)
94.9

(GlcA)
93.6

(Xyl)
98.3

(L-IdoA)
95.4

Red numbers indicate the accuracy in per cent.
*Neu5Ac represents the most common form of sialic acid.

plot of normalized signal peak frequency (that is absolute frequencies divided by the frequency at the highest concentration, values of which are listed in Supplementary Table 7) versus sample concentration. The error bars represent the s.d. on 3–5 repeated measurements. The data were well fitted by a Langmuir function ($R^2 = 0.976$) from which a dissociation constant, $K_d = 0.74 \pm 0.25 \, \mu M$ was obtained (inset in Fig. 2, and see Supplementary Methods for more details). This reflects an increased affinity of the sugar molecule binding to ICAs by forming a triplet confined in the tunnel gap. We used surface plasmon resonance to measure $K_d$ for the adsorption of $\alpha$-$^M$Glu on an ICA monolayer, finding a value of 4 mM (Supplementary Methods and Supplementary Table 9). One important contribution to the enhanced affinity comes from the simultaneous interaction of the sugar molecule with a fixed pair of ICA molecules in the tunnel gap. Assuming that entropy changes were same for both ICA molecules, and equal to that for a single binding event, the adsorption free energy would be doubled on binding at two sites. Consequently, $K_d$ for the two-site binding can be as large as the square of the value for binding by one site, that is $(4 \, mM)^2$ or 16 µM. This is still 20 times larger than the observed 0.74 µM. The electric field in the gap exceeds $10^8 \, V \, m^{-1}$, which may contribute to this enhancement by increasing retention of bound molecules. The additional free energy owing to the molecular dipole (calculated to be 1.17 Debye for $\alpha$-$^M$Glu) can be up to 20% of thermal energy and this will extend the retention of trapped molecules. The same field may also result in dielectrophoretic concentration of molecules in the vicinity of the junction, kinetically enhancing the effective on-rate. These factors combined lead to a remarkably enhanced capture affinity. The count rate at a given concentration is quite reproducible (Supplementary Table 7), and Fig. 2 shows that the RT measurement has a dynamic range that is greater than four orders of magnitude. A few counts per minute are obtained with concentrations as low as 10–100 nM, a signal frequency reduced by only a factor of two once the chemically-insensitive signals are filtered out. Since each of the remaining signal spikes can be

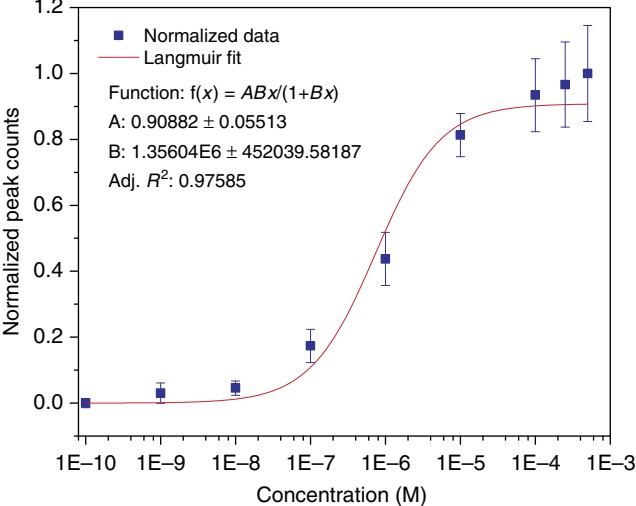

Function: $f(x) = ABx/(1+Bx)$
A: $0.90882 \pm 0.05513$
B: $1.35604E6 \pm 452039.58187$
Adj. $R^2$: 0.97585

**Figure 2 | Count rates as a function of concentration.** Plot of normalized RT counting rates vs concentration of $\alpha$-$^M$Glu for trapping the analyte in an RT gap functionalized with ICA molecules and a fit to a Langmuir isotherm.

assigned with the predictive accuracies, a few minutes of data acquisition allows for classification of solutions down to a sub-10-nM range. This quantitative ability offers a way to count the number of signal spikes of a given character so as to quantify the relative amount of a given isomer. In comparison, mass spectroscopy, which is not inherently quantitative, requires additional techniques, such as isotope labelling for quantification of sample concentrations. The present experiments used a large volume (200 µl) of sample to fill the liquid cell of the scanning tunnelling microscope. However, micron-scale solid-state tunnel junctions are becoming available[22], and it is practical to use microliter volumes with such small devices. With tens of microliter volumes and 10–100 nM concentrations, only $10^{-13}$

to $10^{-14}$ moles of sample will be required for RT measurements. It should also be pointed out that the frequency of capture does depend on the analyte. For example, α-$^M$Glu is captured at $1.8 \times$ the frequency of β-$^M$Glu, and galactose three times more frequently than glucose (Supplementary Table 8). There is some run to run variation, but these ratios are reproducible to within about ±15% and can be corrected for by calibration.

## Discussion

In the present work, we have demonstrated the power of RT to discriminate between various individual saccharides at a single-molecule level. Thus, RT can be used to detect some cancer markers like Tn antigen (GalNAcα1-O-Ser/Thr)[19] and O-GlcNAcylation[23] in a protein sample. As is the case for IM–MS (ref. 9), identifying a broad range of analytes will require construction of the appropriate databases, and this will prove to be a more demanding task for RT because IM–MS also collects independent measurements of ion-masses, providing an independent check of signal assignments. Clearly, neither technique could analyse the vast number of possible isomers in a long oligosaccharide at the present time[24]. However, RT is clearly capable of providing a key component of a single-molecule sequencing system for linear oligosaccharides, a presently almost intractable problem. It has been proposed that RT electrodes incorporated into a nanopore can be used for sequencing linear heteropolymers such as DNA (ref. 25). Since GAGs can effectively be translocated through solid-state nanopores of <3 nm diameters (unpublished results), combining a RT junction with a nanopore (as we are doing for DNA sequencing) to present the individual mono- or disaccharides to the electrodes sequentially, may allow the compositional sequence of linear oligosaccharides to be read out directly. GAGs represent an important class of unbranched polysaccharides for which the compositional sequences are very difficult to obtain by current methods, but with a compositional variability that is small enough to make use of a data-base practical for sequencing[26]. Since the nanopore has no limit on length of molecules that are translocated, the RT-nanopore technique may prove to be an important adjunct to IM–MS with a unique capability for analysis of very small amounts of sample and also enabling the sequence of linear oligosaccharides to be read directly. RT nanopores offer the additional benefit of capturing molecules at the edge of the pore, allowing much bigger (tens of nanometres) pores to generate signals. Means for feeding polymers into large pores sequentially are currently being investigated.

## Methods

**Functionalization of STM probes.** The functionalization of STM probes followed a procedure developed in our lab[27]. First, the STM probes were made from a 0.25 mm Pd wire (California Fine Wires) by AC electrochemical etching in a mixed solution of 36% hydrochloric acid and ethanol (1:1), and then coated with a high-density polyethylene film, having an open apex with a few tens of nanometres in diameter. The insulated probes were gently cleaned by absolute ethanol (200 proof), dried with a nitrogen flow, immersed in an ethanolic solution of ICA (0.5 mM, degassed by argon) for 20 h at room temperature, and then gently rinsed with ethanol and dried with nitrogen. All the STM probes were freshly prepared before each experiment.

**Functionalization of palladium substrates.** The substrate was prepared by depositing 100 nm thick palladium on a 750 μm silicon wafer coated with a 10 nm titanium adhesion layer using electron-beam evaporator (Lesker PVD 75). Palladium substrates were functionalized with ICA in the same way as we did for the STM probes and characterized with various physical and chemical tools (Supplementary Fig. 2; Supplementary Methods).

**RT measurements.** The RT measurements were carried out in PicoSPM (Agilent Technologies) with customized LabView interface for data acquisition. Each tip was tested to ensure that the current leakage was <1 pA in a PB solution at a

500 mV bias. The current set point was 4 pA, corresponding to a gap size of about 2.5 nm between the tip and substrate[28]. The tip was approached to substrate under 1.0 integral and proportional gain servo control (Supplementary Fig. 3). The surface first was scanned to ensure that the tip is not over-coated by high-density polyethylene, so electrodes were a good condition for the RT measurement. After a clear grain structure on the Pd substrate was obtained, the tip was withdrawn about 1 μm and the bias was turned off to avoid possible damage to the ICA layer during the 2-h instrument stabilization. Subsequently, the tip was re-engaged, and the integral and proportional gains were set to 0.1 (see Supplementary Methods for characterization of the servo response).

A phosphate buffer (1 mM, pH 7.4) was used as a control for the RT measurements. Before every measurement, a spectrum of the buffer solution was recorded, and noise spikes were observed only very rarely. Note that these signals are much cleaner (that is, very low peak count rates) compared with our original work[18]. Supplementary Fig. 4 illustrates the RT spectra measured with different electrodes in different solutions. Supplementary Table 6 lists statistical data of electrical spike frequencies in different solutions measured with both bare and ICA functionalized electrodes.

**Data analysis.** To perform the SVM analysis, we first converted the measured electronic signals (current versus time) to sets of numeric values for the features, followed by data normalization, feature selection to remove strongly correlated features, and filtering to remove noise signals as described below.

The baseline of tunnelling current (4 pA) was shifted to zero, and all the current spikes with peak amplitude 15 pA above the baseline were characterized as signals. This amplitude was chosen empirically as being well above spurious electronic noise from the control system of the STM and the environment. This cut off also reduced the fraction of events that were found to be insensitive to the chemistry of the analytes (details of noise filtering are given below). Signal spikes were counted as being in a cluster by convoluting each peak with a Gaussian window of 4,096 data-point full width and unit height (the duration of each collected data point is 20 μs). The convoluted Gaussian traces were summed and a cluster was assigned when the sum exceeds 0.1. In the cluster, features were derived from all the spikes including those with amplitude below 15 pA. The primary features were defined in time domain and the secondary features in frequency or other domains (The definition of domains is illustrated in Supplementary Fig. 6). Each of the peaks and clusters was Fourier transformed and down-sampled to 61 equal frequency bins for clusters and 10 for peaks from zero up to the Nyquist frequency of the instrument (25 kHz in this study) named Peak/Cluster FFT. The feature Peak/Cluster FFT Whole was down-sampled spectrum into 51 bins. The Cepstrum features are the Fourier transform of the power spectrum of the clusters and they were downsampled to 61 bins. The definitions of tunnelling data features are described in Supplementary Methods.

To avoid features that span a large numeric range from dominating features that span a small numeric range, all the calculated features were normalized. The mean of each feature was shifted to be zero and the data scaled to make s.d. = 1.

We began with 264 starting features. Features with a correlation coefficient larger than 0.7 were replaced with a representative feature. The feature variation of the repeated experiments and different analytes are calculated by comparing the single feature histogram for one experiment with the accumulated histogram for multiple repeat experiments. The difference between the repeated runs histogram and the accumulated histogram of the given analyte is assigned as in-group fluctuation (variation of the repeats). The difference of the normalized histogram between the possible analyte pairs is out-group fluctuation (variation of the analytes). The features were ranked by the ratio between the in-group fluctuation and the out-group fluctuation, and the low ranked features were dropped. The surviving features were evaluated by the classification accuracy, and an optimized set of features chosen to get the maximum true positive accuracy.

RT signals that were strongly correlated with z-piezo signal and the water signals (no analyte present) were removed from the data pool. To avoid tunnelling distortion by z-piezo servo control, the clusters highly correlated with z-piezo displacement were removed. With the threshold 0.1 for z-piezo correlation coefficient, ∼10% of peaks are rejected by this process. The water class was defined by the control experiment and the corresponding class in analyte RT signals was rejected. The results in Tables 1 and 2 are obtained with ∼50% rejection of peaks by threshold 0.7.

We used the kernel-mode SVM available from ⟨https://github.com/vjethava/svm-theta⟩. The SVM running parameters C and gamma were optimized through cross-validation of randomly selected sub data set. Full details of the SVM (written in Matlab) can be found in the website: https://github.com/ochensati/SVM_DNA_TunnelVision

**RT quantitation of α-$^M$Glu.** A 1.0 mM stock solution of α-$^M$Glu in the phosphate buffer was diluted to various concentrations from 500 μM to 100 pM. For each measurement, an analyte solution (200 μl) was injected into the liquid cell using a syringe attached to a micro filter. After the measurement, the liquid cell and electrodes were rinsed with the phosphate buffer solution (3 ml) through the fluidic channels to obtain a clean control signal. A pair of electrodes was able to carry out three measurements with different concentrations and the measurement at each concentration was repeated at least two times (Supplementary Table 7). The

isotherm absorption data were analysed in software OriginPro 2016 using the Levenberg–Marquardt algorithm for fitting to a Langmuir equation: $f(x) = a \times (b \times x)/(1 + b \times x)$.

**Data Availability.** All data is available from the authors on reasonable request.

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

## Acknowledgements

This work was supported by grant HG006323 from the National Human Genome Research Institute and R21GM118339 from National Institute of General Medical Sciences.

## Author contributions

J.I., H.L. and Y.Z. carried out RT measurements; J.I. and B.A. and S.L. RT data analysis; So.B. and P.Z. synthesis of ICA, NMR and MS experiments and the data analysis; Su.S. characterization of functionalized Pd surfaces; Su.B. and P.Z. surface plasmon resonance experiments and data analysis; X.W. synthesis of D0A4 and D0A6 and mutarotation analysis of glucose by NMR; P.Z. computer modelling; C.B. designed MS experiments and data analysis; S.L. and P.Z. designed the project and wrote the manuscript.

## Additional information

**Competing financial interests:** The authors declare no competing financial interests.

