## [Peer review file · Nature Communications]

PEER REVIEW FILE

Reviewer #3 (Remarks to the Author):

The authors make the following claims in this paper:

- * Polysaccharides (long chains of monosaccharides) can have different epimers (each monosaccharide can have a different orientation around the stereocenter connecting monosaccharide units). This makes differentiating between them using mass spec etc. difficult. RT seeks to address this issue.
- * RT requires < 1 picomoles of the sample while NMR etc. require > 1 mmole.
- * Can be combined with a solid-state nanopore to sequence polysaccharides (not shown).
- * Signals contain information about identity of the trapped molecule.
- * Concentration of the analyte can be quantified through the peak count rate.

This paper seems to be a rather straightforward application of their existing technology to a new set of molecules. We have some questions/concerns.

Comments

- * Figure 1h shows the X-axis marked as Cluster FFT Whole 37 with limits from -25 to 5. The SI describes cluster FFT whole as "the normalized power spectrum of the cluster, denoised with Wiener filtering and then downsampled into 51 frequency bands in which the bin sizes are spaced...". However, this does not indicate what the X-axis in the figure is. Clarification on this issue would be desirable. The same is applicable for Peak FFT 9 in Figure 1i.
- * The authors mention in the Methods section for Data analysis that "Each of the peaks and clusters was Fourier transformed and down-sampled with 410 Hz width corresponding to 61 equal frequency bins...". However, in the SI section on Data analysis, Peak/Cluster FFTs are mentioned to be downsampled to 9 bins and Cluster FFT Whole is mentioned to be downsampled to 51 bins. This needs to be clarified.
- * The data analysis involves calling spikes with amplitude > 15 pA as signals. Justification on why 15 pA was chosen (by comparing to baseline noise, for example) would be helpful. Also, Fig. S4b,c show that spikes of amplitudes ~25 pA were visible with bare electrodes. Does this mean that some of the actual data would involve these spurious spikes as well? If so, what are its implications? The Noise

filtering subsection of Methods describes a technique for removing water signals (with no analyte present), but does not address the issue of signals being generated with bare electrodes in solutions with analytes.

* The authors mention that the proposed technique could possibly be extended to detecting linear oligosaccharide chains as well as sequence DNA by placing the reader junctions inside a nanopore. From my reading of the work, the authors needed a precise 2.2 nm tunnel width in order to obtain their results. Based on my review of nanopore literature, reproducibly creating such small diameter pores remains a challenging task. The authors could, perhaps, describe what the impact of an inaccurate pore diameter for their technique.

* The authors report that they observed an average peak count of 2.015 peaks/sec at 100 μ M concentration for α -Glu. The data from Figure 1f shows many tens of peaks in a 500 ms window. Does this mean that the average count was calculated from only a few such bursts over the 40 minute duration of the experiment (Methods -> RT Measurements). If so, then does this imply that the classification data is generated based on the interactions of a few copies of the molecule? This should be explained better.

* Figure 1h shows the X-axis marked as Cluster FFT Whole 37 with limits from -25 to 5. The SI describes cluster FFT whole as "the normalized power spectrum of the cluster, denoised with Wiener filtering and then downsampled into 51 frequency bands in which the bin sizes are spaced...". However, this does not indicate what the X-axis in the figure is. Clarification on this issue would be desirable. The same is applicable for Peak FFT 9 in Figure 1i.

We thank the referee for pointing out this problem. We named the data using the physical parameters from which they were derived, but the process of normalizing and rescaling for SVM analysis leads to hard-to-interpret values in these plots. On reflection, we think it best to avoid presenting such derivative data so early in the paper (the technical expert can refer to the SI). Therefore, we have substituted raw data for two straightforward signal features that show some indication of separation in their distributions (new Figures 1h and i). To illustrate how the distribution of simultaneous occurrence of signal features better separates data, we have changed to a plot made using a principle component analysis (PCA) that shows a rather simple partitioning of the data in a 3D plot (Figure 1j). Though these vectors themselves have no simple physical interpretation (they are normalized and dimensionless) the technique is standard in the literature and we list the composition of each vector in the SI.

* The authors mention in the Methods section for Data analysis that "Each of the peaks and clusters was Fourier transformed and down-sampled with 410 Hz width corresponding to 61 equal frequency bins...". However, in the SI section on Data analysis, Peak/Cluster FFTs are mentioned to be downsampled to 9 bins and Cluster FFT Whole is mentioned to be downsampled to 51 bins. This needs to be clarified.

Thank you for pointing this discrepancy out. The methods section has been corrected as follows:

Each of the peaks and clusters was Fourier transformed and down-sampled to 61 equal frequency bins for clusters and 10 for peaks from zero up to the Nyquist frequency of the instrument (25 kHz in this study) named Peak/Cluster FFT. The feature Peak/Cluster FFT Whole was down-sampled spectrum into 51 bins. The Cepstrum features are the Fourier transform of the power spectrum of the clusters and they were downsampled to 61 bins. The definitions of tunneling data features are described in Section 8 of SI.

* The data analysis involves calling spikes with amplitude > 15 pA as signals. Justification on why 15 pA was chosen (by comparing to baseline noise, for example) would be helpful. Also, Fig. S4b,c show that spikes of amplitudes ~25 pA were visible with bare electrodes. Does this mean that some of the actual data would involve these spurious spikes as well? If so, what are its implications? The Noise filtering subsection of Methods describes a technique for removing water signals (with no analyte present), but does not address the issue of signals being generated with bare electrodes in solutions with analytes.

Feature extraction: the baseline of tunneling current (4 pA) was shifted to zero and all the current spikes with peak amplitude 15 pA above the baseline were characterized as signals. This amplitude was chosen empirically as being well above spurious electronic noise from the control system of the STM and the environment. This cut off also reduced the fraction of events that were found to be

insensitive to the chemistry of the analytes (see *Noise Filtering* below).

* The authors mention that the proposed technique could possibly be extended to detecting linear oligosaccharide chains as well as sequence DNA by placing the reader junctions inside a nanopore. From my reading of the work, the authors needed a precise 2.2 nm tunnel width in order to obtain their results. Based on my review of nanopore literature, reproducibly creating such small diameter pores remains a challenging task. The authors could, perhaps, describe what the impact of an inaccurate pore diameter for their technique.

We have added at the end of the main text:

RT nanopores offer the additional benefit of capturing molecules at the edge of the pore, allowing much bigger (tens of nm) pores to generate signals. Means for feeding polymers into large pores sequentially are currently being investigated.

* The authors report that they observed an average peak count of 2.015 peaks/sec at 100 μ M concentration for alpha-Glu. The data from Figure 1f shows many tens of peaks in a 500 ms window. Does this mean that the average count was calculated from only a few such bursts over the 40 minute duration of the experiment (Methods -> RT Measurements). If so, then does this imply that the classification data is generated based on the interactions of a few copies of the molecule? This should be explained better.

Further explanation is now added on page 7:

Signals appear in bursts, and strong correlations between signal features in a given cluster (see *Feature Extraction* in Methods) suggest that each cluster reflects structural fluctuations within one particular capture geometry for one molecule. Average count rates (Table S7) are much lower than the count rate within a signal cluster, so the number of distinct molecules sampled in a given run is likely much smaller than the total number of peaks measured.

REVIEWERS' COMMENTS:

Reviewer #3 (Remarks to the Author):

The authors have satisfactorily addressed all the comments that had been noted previously. While the work shows promising results and performs better than existing techniques, the inherent combinatorial complexity of the problem remains, which the authors themselves accept. The authors could experimentally back their claim that the technique could detect relative concentrations of different target molecules in future work.

The SI has a caption for figure S3 and references to the figure in the text but is missing the actual figure.

This work is a valuable addition to the field of recognition tunneling.

Supplementary Figure 3 has been reinserted